# Factors affecting the implementation of evidence-based Progressive Tinnitus Management in Department of Veterans Affairs Medical Centers

Tara L. Zaugg[1]*, Emily J. Thielman[1], Kathleen F. Carlson[1,2,3], Anaïs Tuepker[2,4], Christine Elnitsky[5], Karen L. Drummond[6,7], Caroline J. Schmidt[8], Summer Newell[2], Christine Kaelin[1], Christie Choma[1,2], James A. Henry[1,9]

1 VA RR&D National Center for Rehabilitative Auditory Research (NCRAR), VA Portland Health Care System, Portland, Oregon, United States of America, 2 VA HSR&D Center to Improve Veteran Involvement in Care (CIVIC), VA Portland Health Care System, Portland, Oregon, United States of America, 3 Oregon Health & Science University – Portland State University School of Public Health, Portland, Oregon, United States of America, 4 Division of General Internal Medicine and Geriatrics, Oregon Health & Science University (OHSU), Portland, Oregon, United States of America, 5 College of Health and Human Services, University of North Carolina at Charlotte, Charlotte, North Carolina, United States of America, 6 VA Center for Mental Healthcare and Outcomes Research, Central Arkansas Veterans Healthcare System, North Little Rock, Arkansas, United States of America, 7 Department of Psychiatry, University of Arkansas for Medical Sciences, Little Rock, Arkansas, United States of America, 8 Yale School of Medicine, Department of Psychiatry, New Haven, Connecticut, United States of America, 9 Department of Otolaryngology – Head & Neck Surgery, Oregon Health & Science University (OHSU), Portland, Oregon, United States of America

* tara.zaugg@va.gov

**Data Availability Statement:** All relevant data are within the paper and its Supporting information files.

## Abstract

### Purpose

Progressive Tinnitus Management (PTM) is an evidence-based interdisciplinary stepped-care approach to improving quality of life for patients with tinnitus. PTM was endorsed by Department of Veterans Affairs (VA) Audiology leadership in 2009. Factors affecting implementation of PTM are unknown. We conducted a study to: 1) estimate levels of PTM program implementation in VA Audiology and Mental Health clinics across the country; and 2) identify barriers and facilitators to PTM implementation based on the experiences of VA audiologists and mental health providers.

### Method

We conducted an anonymous, web-based survey targeting Audiology and Mental Health leaders at 144 major VA facilities. Quantitative analyses summarized respondents' facility characteristics and levels of program implementation (full PTM, partial PTM, or no PTM). Qualitative analyses identified themes in factors influencing the implementation of PTM across VA sites.

**Funding:** This work was supported by the Department of Veterans Affairs, Health Services Research and Development (HSR&D) Service Quality Enhancement Research Initiative (QUERI) through a Rapid Response Project mechanism (RRP 13-440/IRB3295) (JAH). https://www.queri. research.va.gov The funders had no role in study design, data collection and analysis, decision to publish, or preparation of the manuscript.

**Competing interests:** The authors have read the journal's policy and have the following conflicts: JH, TZ, and CS were part of the research team that originally developed Progressive Tinnitus Management. TZ, ET, KC, AT, KD, SN, CK, CC, and JH are currently affiliated with the Department of Veterans Affairs. The views expressed in this article are those of the authors and do not necessarily reflect the position or policy of the Department of Veterans Affairs or the United States government. The disclosure does not alter our adherence to PLOS ONE policies on sharing data and materials.

## Results

Surveys from 87 audiologists and 66 mental health clinicians revealed that few facilities offered full PTM; the majority offered partial or no PTM. Inductive analysis of the open-ended survey responses identified seven factors influencing implementation of PTM: 1) available resources, 2) service collaboration, 3) prioritization, 4) Veterans' preferences and needs, 5) clinician training, 6) awareness of (evidence-based) options, and 7) perceptions of scope of practice.

## Conclusion

Results suggest wide variation in services provided, a need for greater engagement of mental health providers in tinnitus care, and an interest among both audiologists and mental health providers in receiving tinnitus-related training. Future research should address barriers to PTM implementation, including methods to: 1) improve understanding among mental health providers of their potential role in tinnitus management; 2) enhance coordination of tinnitus-related care between health care disciplines; and 3) collect empirical data on Veterans' need for and interest in PTM, including delivery by telehealth modalities.

## Introduction

The United States (US) Department of Veterans Affairs (VA) consistently reports tinnitus as the most prevalent service-connected disability, with over 1.9 million Veterans service connected for tinnitus in 2018 (Veterans Benefits Administration, Annual Report Fiscal Year 2018). People with tinnitus experience ringing, humming, buzzing, or other noises in their ears or head in the absence of an external sound. In most cases, the perception of the tinnitus sound cannot be permanently changed [1]. People who are negatively impacted by tinnitus commonly report sleep disturbance, difficulty concentrating, and disruptive emotional reactions [2]. These functional effects vary widely in severity, ranging from being a mild nuisance to being completely disabling.

Although there is no cure for tinnitus, therapies exist to address its functional effects. Progressive Tinnitus Management (PTM) is an interdisciplinary (audiology and mental health), stepped-care program that offers a structured approach to providing care for tinnitus [3]. The PTM program includes educational materials for both patients and clinicians, tools for clinicians to assess the impact of tinnitus and to inform decisions regarding care, and integrated materials to enhance the coordination of care across disciplines. PTM was originally developed by VA researchers as a program to benefit Veterans in the context of VA-provided health care. Since its development, PTM programs have also been implemented in military health care facilities [4] and some clinics serving a general US population (N. Bowen-Hicks, personal communication, November 5, 2019).

The general role of audiology in providing care for tinnitus includes referring to physicians for medical assessment as indicated, identifying comorbid hearing loss, addressing functional effects of hearing loss, providing educational counseling, and working with individuals to develop appropriate expectations for using sound to improve quality of life with tinnitus. PTM was initially developed to provide structured guidance on fulfilling the above stated role of audiologists in tinnitus [5, 6]. However, even in an audiology context, mental health issues arose; for example, patients described suicidal ideation and intense emotions associated with

their tinnitus. Therefore, it became clear that collaboration with mental health providers was necessary in working with some patients distressed by their tinnitus.

It is widely recognized that mental health symptoms and disorders are often comorbid with tinnitus [7]. Although the causal direction of this association is not well understood, it is likely that bothersome tinnitus can exacerbate mental health symptoms while, conversely, mental health disorders can have a negative effect on the perception of—and ability to cope with—tinnitus. It is also possible that tinnitus could result in the genesis of new mental health symptoms for those with no preexisting mental health issues. According to systematic reviews, the strongest research evidence for improving quality of life and reducing depressive symptoms for individuals with tinnitus is for Cognitive Behavioral Therapy (CBT), a psychological therapy centered around modifying thoughts and behavioral responses to distress [8–10]. Practical needs combined with prevailing research evidence led to incorporating CBT into PTM and coordinating with mental health clinicians to administer the intervention. CBT coping strategies, along with education on using sound to help with tinnitus, comprise the coping skills that are offered within the PTM framework. The five levels of PTM range from initial referral and basic education to ongoing individualized support, with each individual patient progressing only as far as necessary. The progressive levels and components of PTM are depicted in Fig 1.

PTM has been shown in two randomized controlled trials (RCTs) to reduce tinnitus distress and improve life satisfaction for tinnitus sufferers in both clinical and controlled research settings, including through remote administration by telephone [3, 11]. The first RCT evaluated PTM as a clinically-embedded program. Results showed statistically significant (though modest in scale) improvements in participants who learned PTM coping skills in a group setting, including reductions in the negative impact of tinnitus on functioning and increased self-efficacy [3]. In the second RCT, the progressive levels of PTM were modified for one-on-one administration over the telephone [11]. In this individual-based version of PTM, in which coping skills were taught during separate telephone appointments with an audiologist and a mental health provider, considerable improvements were seen in tinnitus outcomes following participation. These improvements were evident as early as 3 months post-baseline and were sustained through at least 12 months. Moderate improvements were also seen in scales measuring anxiety, depression, and self-efficacy for managing symptoms.

Progressive Tinnitus Management has been endorsed as a standard of care by national VA leadership since 2009 [12]. The VA is the largest integrated health care system in the US with numerous sites throughout the country. Each site has multiple disciplines working within the same system, forming an ideal setting for implementing interdisciplinary care for tinnitus. VA has a documented interest in collaborative care models [13–15], supporting programs such as Primary Care—Mental Health Integration (PCMHI) and Patient Aligned Care Teams (PACTs), designed to emphasize patient-centered, team-based care and to increase access to commonly required services. However, despite the enormous need for tinnitus care among Veterans, strong support from VA leadership at the national level, and clinical environments conducive to multidisciplinary care, implementation of PTM across the VA has been slow and inconsistent [16].

Identifying factors that drive or inhibit PTM uptake and clinical delivery is vital for the success of future efforts to increase the spread of PTM across the VA or other health care systems. The objectives of the present study were to: (1) describe levels of PTM program implementation in VA audiology and mental health clinics across the country; and (2) identify factors affecting PTM program implementation based on the experiences of VA audiology and mental health clinicians at sites that have fully, partially, or not implemented PTM.

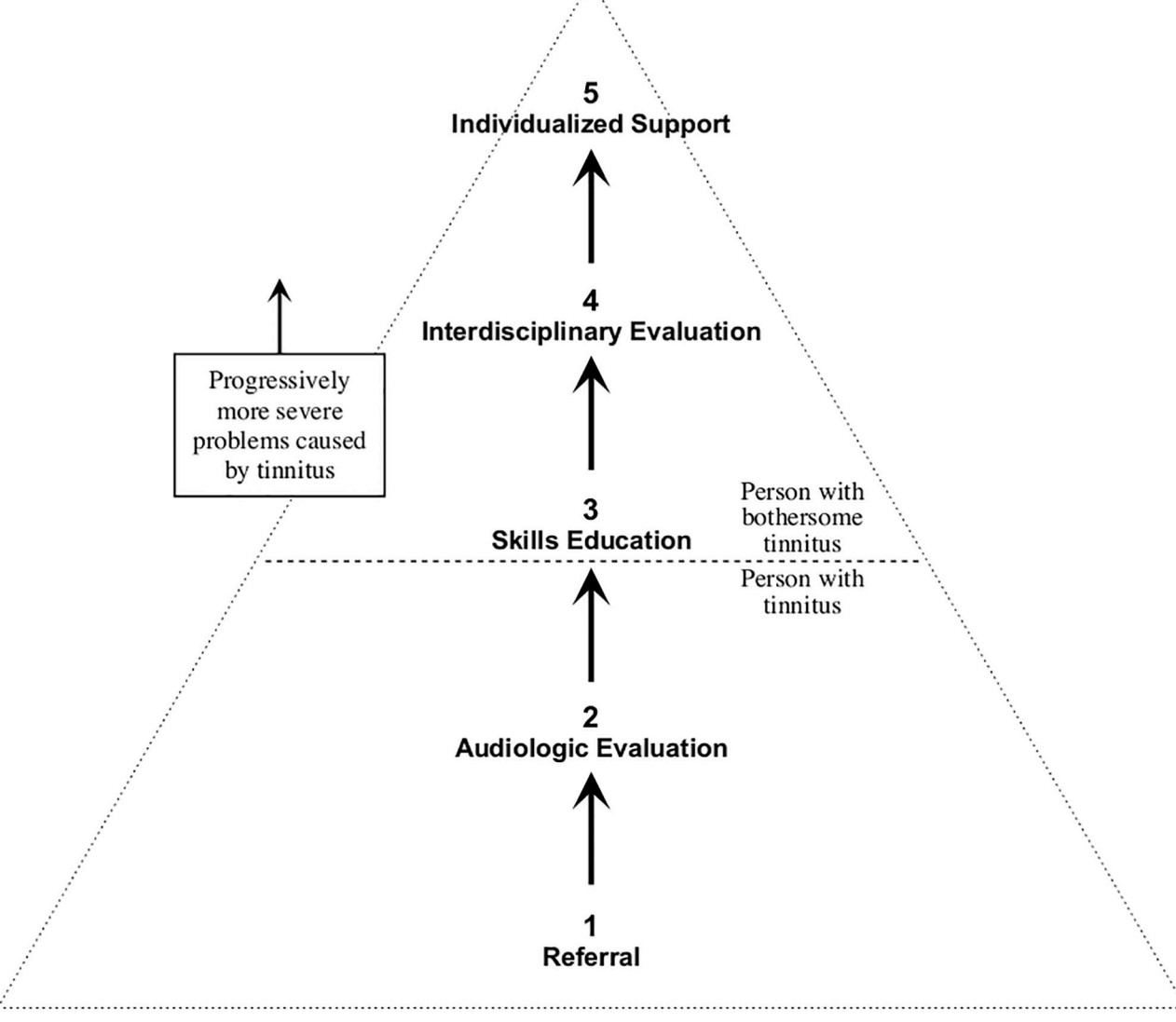

**Fig 1. Tinnitus pyramid.** The pyramid depicts the entire population of adults with tinnitus. The wide base of the pyramid represents those not particularly bothered by their tinnitus, as the majority of adults with tinnitus fall in this area. Severity increases as you move upwards, with the tip of the pyramid showing the small fraction of individuals who are most severely affected. The five levels of Progressive Tinnitus Management (PTM) are overlaid on the pyramid to reflect the progression from basic to more involved levels of care as needed, based on the severity of the problem with tinnitus.

## Methods

### Overview

This study collected data via an anonymous, web-based survey of audiology and mental health leaders/clinicians across the VA. The survey was conducted in 2015 using the Research Electronic Data Capture (REDCap) web application behind the VA firewall. The conduct of this study was approved by the VA Portland Health Care System (VAPORHCS) Institutional Review Board (IRB).

## Study population

The target population for this survey was audiology and mental health service chiefs (or similarly knowledgeable supervisors/clinicians) at all major VA medical center sites nationwide that provided both audiology and mental health services. We identified 144 sites with both clinic types in 2015. Our aim was to collect one audiology and one mental health survey from each site, in order to describe the implementation of PTM services on a site level across the country. While this sampling strategy would not include all clinicians potentially engaged in delivering PTM, we believed it would focus our sample on those individuals most involved in PTM delivery, whose responses would be most informative toward our goal of understanding and improving levels of PTM implementation in the VA. Contact information for survey recipients was assembled using a master list provided by the respective national service offices (Audiology and Speech Pathology, and Mental Health) in VA's Central Office, supplemented with information from personal contacts, an internal VA global address book, and the internet. Because individual VA site organizational structure varies, emails to the identified service chiefs/supervisors requested that they or a clinician knowledgeable about tinnitus clinical services at their site complete the anonymous survey. The invitation acknowledged the study team's collaboration with the national service offices and the VA Health Services Research and Development (HSR&D) Service Quality Enhancement Research Initiative (QUERI).

## Informed consent

VA clinicians were asked to give consent to participate in research prior to completing the survey by indicating 'yes' to the first survey question, which included a description of the research. If the individual did not choose 'yes' to this question, the survey was terminated. Although the survey was anonymous, participants were asked to report their VA station/facility number and their Veterans Integrated Service Network (VISN) number, an administrative indicator of the geographic region in which each VA station/facility is located. These numbers were the only "identifying" information retained.

## Survey development and data collection

Separate audiology and mental health surveys were designed by the study team and vetted with clinical and operations partners prior to fielding (S1 and S2 Appendices). Survey questions were developed to cover several domains of interest, including descriptive information about the clinician, clinic, facility, patient population, referrals, and tinnitus services available. Questions addressing an additional domain, namely training preferences, were added after being proposed by employee education partners within VA. The surveys were reviewed by the research team and colleagues within VA HSR&D for clarity, flow, and accessibility.

The audiology and mental health surveys contained nearly all of the same questions, though response options were structured according to discipline. The first section of the survey asked about participants' facility characteristics and a second section focused on level of tinnitus management program implementation at their facility, including tinnitus patient characteristics and service use. The first section included five nominal, three ordinal, and one Likert-type question regarding characteristics and five interval, two nominal, and one Likert-type question regarding provider training. Additionally, the first section asked two open-ended questions regarding whether the respondent believes their clinic should provide tinnitus services and the respondent's confidence in providing those services. The second section included four

nominal, six ordinal, two interval, and one Likert-type question as well as one open-ended question asking the respondent to identify the main challenges to using tinnitus management services at their facility. The surveys were kept to a length that could be completed in 30 minutes or less.

The final audiology and mental health surveys were programmed into REDCap, with web links generated for each. These survey links were disseminated in invitation emails to the lists of 144 audiology service chiefs and 144 mental health service chiefs representing each VA facility with these services. The survey web page was open for approximately one month. After the initial invitation, weekly reminder emails were sent for four weeks.

## Data analyses

Although our original intent was to link audiology and mental health survey responses from the same facility using the site code, this was not possible in most cases, as there was little overlap in site codes between audiology and mental health respondents. Thus, responses from the audiology and mental health surveys are presented separately.

**PTM categories.** Each survey respondent's site was categorized as providing "full PTM," "partial PTM," or "no PTM." Our operating definition of a full PTM program required at least two PTM skills education sessions conducted by an audiologist and two sessions conducted by a mental health provider (either in a group setting or one-on-one).

To be categorized as providing partial PTM, a site must have reported a commitment of resources and concerted effort to teach PTM coping skills. The minimum requirements were the provision of coping skills offered in a group or individual setting by at least one discipline. Any site offering care by both disciplines, but with only a single session provided by one or both disciplines, was also categorized as partial PTM.

All other responses that did not fit into either the full PTM or partial PTM categories were classified as no-PTM sites. These no-PTM sites may have offered some tinnitus care, but the care did not meet the definition of full or partial PTM.

**Quantitative and qualitative analyses.** Closed-ended survey responses were analyzed descriptively using SAS 9.4 software. Descriptive statistics (e.g., frequencies) were calculated for selected survey questions, stratified by survey type (audiology versus mental health). Some response options were collapsed to minimize zero-count cells. Where available, write-in explanations for an 'other' response were assessed and additional categories added when sufficient write-ins gave the same or similar responses. PTM category was cross-tabulated with other key variables and Chi-square analyses of these cross-tabulations were performed. Response categories were combined where possible to reduce low or zero count cells, in order to meet the criteria for a Chi-square analysis.

For analysis of the open-ended survey items, two team members with qualitative research expertise (authors AT, CE) double-coded responses using an initial coding scheme based on domains of *a priori* interest suggested by implementation science literature, in particular the Consolidated Framework for Implementation Research (CFIR) [17]. Planned, iterative discussion of this preliminary coding then proceeded to develop inductive themes with the input of the larger team, including analysts with PTM program expertise (authors TZ, JH). This took place over at least five formal sessions, during which themes were refined and differences in interpretation of responses were resolved by consensus in alignment with accepted practices to enhance qualitative rigor [18]. All resulting themes are reported; while the coverage or overall prevalence of themes is not reported, our analysis was attentive to differences in the nuances of themes as they surfaced in responses of mental health versus audiology providers.

## Results

Surveys were completed by 87 audiology chiefs/clinicians and 66 mental health chiefs/clinicians, representing approximately 60% and 46% of recruited VA facilities, respectively. Almost all participants (95%) provided responses to one or more open-ended questions.

Eighty-six percent of audiology chiefs/clinicians and 62% of mental health chiefs/clinicians reported their clinics to be located within VA hospitals; the remainder were located in community-based outpatient clinics. The majority of audiology respondents were clinical audiologists (82%). Of the others, 15% identified themselves as service chiefs or supervisors, and 2% as audiology trainees. Among mental health respondents, 36% were clinical psychologists, 12% were clinical social workers, 15% were service chiefs or supervisors, 17% were psychiatrists, 12% were registered nurses or nurse practitioners, and 3% were trainees (the remaining 5% reported 'Other'). Further characteristics of survey respondents and their sites are presented in Table 1.

**Table 1. Descriptive characteristics of survey respondents.**

|  | Audiology Survey | | Mental Health Survey | |
|---|---|---|---|---|
|  | N | % | N | % |
| *Number of Responses* | 87 | | 66 | |
| *Hospital facility* | | | | |
| *Hospital* | 75 | (86%) | 41 | (62%) |
| *Outpatient clinic* | 8 | (9%) | 23 | (35%) |
| *Other* | 2 | (2%) | 1 | (2%) |
| *Missing* | 2 | (2%) | 1 | (2%) |
| *Role at facility* | | | | |
| *Supervisor or Service Chief* | 13 | (15%) | 10 | (15%) |
| *Clinical Audiologist/Psychologist* | 71 | (82%) | 24 | (36%) |
| *Clinical Social Worker* | . | . | 8 | (12%) |
| *Trainee* | 2 | (2%) | 2 | (3%) |
| *Other* | 1 | (1%) | 22 | (33%) |
| *How many FTE[a] audiologists/MH[b] providers employed at your clinic?* | | | | |
| *1–5* | 37 | (43%) | 14 | (21%) |
| *6–10* | 33 | (38%) | 9 | (14%) |
| *11–15* | 12 | (14%) | 2 | (3%) |
| *16–20* | 2 | (2%) | 5 | (8%) |
| *More than 20* | . | . | 30 | (45%) |
| *Missing* | 3 | (3%) | 6 | (9%) |
| *Patient volume per day, per FTE[a] clinician* | | | | |
| *Fewer than 5* | . | . | 3 | (5%) |
| *5–7* | 19 | (22%) | 29 | (44%) |
| *8–10* | 59 | (68%) | 15 | (23%) |
| *11–13* | 6 | (7%) | 9 | (14%) |
| *More than 13* | . | . | 4 | (6%) |
| *Missing* | 3 | (3%) | 6 | (9%) |

[a]FTE = full-time equivalent.
[b]MH = mental health.

## Provision of tinnitus services (or PTM)

Respondents reported a wide variety of tinnitus services across their respective sites. The majority of audiology participants (83%) indicated that tinnitus services were available in their clinics, whereas few mental health participants (14%) reported the availability of tinnitus services in their clinic. Among the audiology participants, about 30% were classified as sites that offered full PTM, while 8% of mental health participants were categorized as offering full PTM. These and other results relating to the provision of care are presented in Table 2.

In response to a question asking, "Do you feel that your clinic should provide tinnitus-management services to Veteran patients?" 100% of audiology respondents and 70% of mental health respondents replied, "Definitely yes" or "Probably yes." Similarly, in response to a question asking, "How confident are you that your clinic can provide tinnitus-management services to Veteran patients?" fewer of the mental health respondents reported having confidence in their clinic's ability to provide tinnitus services (40% very confident or somewhat confident) than audiology respondents (95% very confident or somewhat confident).

## Clinician interest in training

Although audiology respondents reported higher levels of readiness to provide PTM than mental health respondents, large proportions of both types of respondents (90% of audiology and 66% of mental health) indicated they would be interested in receiving training in tinnitus management if supported by a supervisor (Table 3). Fifty-eight percent of audiology survey

**Table 2. Provision of tinnitus care at respondents' sites.**

| | Audiology Survey | | Mental Health Survey | |
|---|---|---|---|---|
| | N | % | N | % |
| *Number of Responses* | 87 | | 66 | |
| *PTM[a] Category* | 26 | (30%) | 5 | (8%) |
| *Full PTM[a]* | | | | |
| *Partial PTM[a]* | 17 | (20%) | 1 | (2%) |
| *No PTM[a]* | 44 | (51%) | 60 | (91%) |
| *Do you provide clinical services for tinnitus?* | | | | |
| *No* | 15 | (17%) | 57 | (86%) |
| *Yes* | 71 | (82%) | 9 | (14%) |
| *Missing* | 1 | (1%) | . | . |
| *Do you feel your clinic should provide tinnitus management services?* | 78 | (90%) | 23 | (35%) |
| *Definitely yes* | | | | |
| *Probably yes* | 9 | (10%) | 23 | (35%) |
| *Probably no* | . | . | 12 | (18%) |
| *Definitely no* | . | . | 8 | (12%) |
| *How confident are you that your clinic can provide tinnitus management services?* | | | | |
| *Very confident* | 51 | (59%) | 6 | (9%) |
| *Somewhat confident* | 32 | (37%) | 20 | (30%) |
| *Barely confident* | 3 | (3%) | 16 | (24%) |
| *Not at all confident* | 1 | (1%) | 23 | (35%) |
| *Missing* | . | . | 1 | (2%) |

[a]PTM = Progressive Tinnitus Management.

**Table 3. Respondents' preferences for receiving tinnitus management training.**

| | Audiology Survey | | Mental Health Survey | |
|---|---|---|---|---|
| | N | % | N | % |
| *Number of Responses* | 87 | | 66 | |
| *Interested in receiving tinnitus management training?* | | | | |
| Definitely yes | 55 | (63%) | 17 | (26%) |
| Probably yes | 19 | (22%) | 23 | (35%) |
| Not sure | 4 | (5%) | 10 | (15%) |
| Probably no | 4 | (5%) | 5 | (8%) |
| Definitely no | . | . | 5 | (8%) |
| Missing | 5 | (6%) | 6 | (9%) |
| *How much time willing to invest in training?* | | | | |
| Up to 1 hour | . | . | 10 | (15%) |
| 1–3 hours | 13 | (15%) | 13 | (20%) |
| 3–5 hours | 16 | (18%) | 6 | (9%) |
| 5–10 hours | 26 | (30%) | 7 | (11%) |
| 11–20 hours | 24 | (28%) | 12 | (18%) |
| None | 1 | (1%) | 10 | (15%) |
| Missing | 7 | (8%) | 8 | (12%) |
| *Which methods of training would interest you?* | | | | |
| Online (N and % Yes) | 57 | (66%) | 35 | (53%) |
| Live training at site (N and % Yes) | 56 | (64%) | 43 | (65%) |
| Live training offsite (w/travel) (N and % Yes) | 42 | (48%) | 16 | (24%) |
| Self-study (N and % Yes) | 37 | (43%) | 16 | (24%) |
| Other (N and % Yes) | 4 | (5%) | 5 | (8%) |
| *How would you prefer to receive clinical supervision?* | | | | |
| Direct in-clinic supervision (N and % Yes) | 57 | (66%) | 41 | (62%) |
| Audio-record with feedback (N and % Yes) | 16 | (18%) | 15 | (23%) |
| Video-record with feedback (N and % Yes) | 19 | (22%) | 8 | (12%) |
| Other (N and % Yes) | 9 | (10%) | 15 | (23%) |

respondents indicated willingness to invest 5–20 hours in training over the course of a year. Mental health survey respondents were more varied, with 35% indicating willingness to invest up to 3 hours of time for training, and 29% indicating willingness to invest 5–20 hours over a year. When asked about training method preferences, only about half of the audiology respondents (48%) and a quarter of mental health respondents (24%) were interested in training offsite requiring travel. Nine mental health providers suggested additional training preferences beyond those listed in the survey; seven of those nine suggestions were one-on-one and/or group telephone calls. Five audiologists suggested training preferences beyond those listed in the survey; three of those suggestions were options that would be compatible with training and ongoing telephone support from a mentor.

## Cross-tabulations by PTM category

Cross-tabulations of PTM category with respondent and/or site characteristics, training preferences and other variables resulted in many zero count cells, particularly for the mental health survey data. As there were only 4 full PTM and 1 partial PTM mental health survey responses, Chi-square analyses could not be run on those data using the PTM category

**Table 4. PTM category by number of full time equivalent (FTE) providers.**

| How many FTE[b] audiologists employed at your clinic? | PTM[a] Category | | | | | | | |
|---|---|---|---|---|---|---|---|---|
| | Full PTM[a] | | Partial PTM[a] | | No PTM[a] | | Total | |
| | N | % | N | % | N | % | N | % |
| 1–5 | 6 | (23%) | 6 | (35%) | 25 | (61%) | 37 | (44%) |
| 6–10 | 14 | (54%) | 7 | (41%) | 12 | (29%) | 33 | (39%) |
| 11–20 | 6 | (23%) | 4 | (24%) | 4 | (10%) | 14 | (17%) |
| Total | 26 | (100%) | 17 | (100%) | 41 | (100%) | 84 | (100%) |

[a]PTM = Progressive Tinnitus Management.

[b]FTE = full time equivalent.

variable. Chi-square analyses of audiology survey responses were also affected by some low or zero count cells, requiring that some response categories be combined.

Among audiology survey responses, there was a statistically significant relationship between PTM category and the number of full-time providers in the clinic (Chi-Square = 10.32, df = 4, p = 0.035), with the no PTM category distributed more toward sites with fewer clinicians, full PTM distributed symmetrically with the majority reporting 6–10 clinicians, and the partial PTM category showing more of a uniform distribution (Table 4). However, there was no significant relationship between PTM category and daily patient volume (Chi-Square = 0.63, df = 2, p = 0.73), nor with hospital facility (Chi-Square = 2.30, df = 2, p = 0.32). Other comparisons of interest could not be assessed due to cells with zero or low counts.

### Factors influencing implementation of tinnitus management services

Qualitative analysis of the open-ended survey responses identified seven thematic factors influencing providers' abilities or desires to implement tinnitus management services. These were: 1) available resources, 2) service collaboration, 3) prioritization, 4) Veterans' preferences and needs, 5) clinician training, 6) awareness of (evidence-based) options, and 7) perceptions of scope of practice. Table 5 describes these factors, with illustrative quotes and attention to differences between professional groups. Audiologists and mental health providers were closely aligned in their perceptions of needed resources and the benefits of service collaboration. Both audiologists and mental health providers mentioned difficulty finding space to hold group sessions, and staff shortages that prevented implementation of a new program. Lack of resources also factored into other themes in responses, such as prioritization (tinnitus is a lower priority given lack of time to address current top priorities). As groups, audiologists and mental health providers identified different Veteran needs and preferences influencing tinnitus management, different clinician training deficits, and different barriers to prioritizing tinnitus management. There were differences between provider groups in the salience of some thematic factors: notably, mental health providers exhibited wide within-group variation in their perceptions of whether tinnitus management fell within their scope, while audiologists expressed that tinnitus was within their scope of practice. Audiologists described variation among their colleagues' awareness of evidence-based options as a factor that could hinder implementation, while mental health respondents were less aware of techniques to address tinnitus altogether.

### Discussion

This survey was the first to evaluate the provision of evidence-based tinnitus care, and barriers and facilitators to PTM implementation, from both audiologists and mental health providers

**Table 5. Factors influencing implementation of Progressive Tinnitus Management (PTM).**

| Factors | Illustrative Audiology Responses | Illustrative Mental Health Responses |
|---|---|---|
| **1. Available resources**: adequate staff time, sufficient space for group sessions, and availability of materials are all needed to offer/expand PTM[a]. | *"Shortage of staff to implement the program"* | *"It would be nice to have these services, but our clinic is so tiny that I doubt we could provide them unless we moved to a larger facility."* |
| | *"Space is an issue here and not the best set-up for groups"* | *"I am confident our staff could provide good tinnitus management; however, we have staff shortages at our clinic and are already stretched thin."* |
| | *"Tinnitus management handbook is not available sometimes."* | |
| **2. Service collaboration**: within audiology, effective mental health outreach is sometimes missing; within mental health, integrated care delivery models and service budgets were noted to facilitate PTM[a] delivery. | *"I do think Mental Health should be involved but we have not achieved that very well here."* | *"I work in PC-MHI[b]. This is a natural place to provide intervention for medical concerns that can be assisted by behavioral interventions."* |
| | | *"To ensure success, it could be important for the tinnitus services to budget for the MH[c] FTE[d] up front as part of a comprehensive care program, rather than budgeting for all. . .except for the MH component."* |
| **3. Prioritization**: audiologists widely viewed tinnitus treatment as a high priority, though sometimes felt facility management prioritized meeting other patient needs first; within mental health, attitudes varied widely on how and when tinnitus management should be prioritized | *"We have seen a marked increase in [patients] whose primary complaint is bothersome to debilitating tinnitus. I want to provide the appropriate care for this rising population for whom this interferes with their lives."* | *"Staff may not recognize this to be as serious a problem as PTSD[f] or other major mental health conditions."* |
| | *"With the number of TBI[e] cases resulting in younger veterans with tinnitus in the absence of hearing loss, it is imperative that we offer assistance to this group. . . ."* | *"I'm not sure we would have the time to add a new program into our system. We are already behind in trying to meet national standards for other services."* |
| | *"It is a need that our veterans have, but we have not been able to expand our services to provide it due to a lack of support from senior management"* | *"Deafness and hearing loss is one of my specializations. . .but I can't say how much attention is paid to this area [by other mental health]."* |
| | | *"Providers are spread very thin. . . .I'm not sure it can take as high a priority as other services. I would like to see this change!!!"* |
| | | *"It is not something that seems to be on the radar"* |
| **4. Veteran preferences and needs**: audiologists frequently noted logistical/practical challenges; mental health providers mentioned possible lack of Veteran awareness about options. | *"Many patients live some distance from the facility"* | *"I'm sure Vets who come to MH[c] appointments don't bring up the issue [tinnitus] even though it is distressing; they may feel there is nothing MH[c] can do."* |
| | *"The two group sessions deter patients from signing up. More likely to come to one group session."* | |
| | *"Veterans do not stay in the program"* | |
| **5. Clinician training**: while common to both groups, the need for training in tinnitus management was more often mentioned by mental health providers, either for themselves or their colleagues. | *"There is only one audiologist who has had tinnitus training."* | *"I'm not trained to manage tinnitus. . .I refer out to audiology"* |
| | | *"I do not believe that anyone in my clinic has been trained in any of these protocols."* |
| | | *"I don't think anyone is specifically trained to do this. However we could conduct services after some training (ie, workshop)."* |
| **6. Knowledge of (evidence-based) options**: mental health providers often were not familiar with any management techniques, while audiologists more often described variation in awareness of the evidence base. | *"Some providers are aware of research and are evidence based in their practice and others avoid addressing tinnitus at all."* | *"Mental health staff are not usually familiar with techniques to address tinnitus"* |
| | *"We all seem to be starting the approach of how to work with tinnitus patients differently."* | *"Not trained to treat medical conditions"* |
| **7. Perception of scope of practice**: audiologists uniformly saw tinnitus management as within their scope though some aspects could be challenging for them; some mental professionals did not see tinnitus management as within their scope of practice. | *"Tinnitus management is part of scope of practice as much as hearing loss."* | *"I do not think general mental health is an appropriate service to provide these services."* |
| | *"Sometimes mental health issues, PTSD[f], etc., impact tinnitus management and Audiologists may feel some issues encountered are out of our scope of practice and/or comfort zone."* | *"Tinnitus is not generally defined as a mental disorder"* |
| | | *"Not sure what mental health can offer"* |
| | | *"Psychiatrists are not trained to provide this type of service. We have an excellent audiology dept here that does a wonderful job providing these services"* |

[a]PTM = Progressive Tinnitus Management.

[b]PC-MHI = Primary Care—Mental Health Integration.

[c]MH = mental health.

[d]FTE = full time equivalent.

[e]TBI = traumatic brain injury.

[f]PTSD = post-traumatic stress disorder.

in VA settings. Our results illuminated thematic factors common across implementation efforts in health care settings, as discussed below. Other approaches to coding our qualitative data might have generated slightly different names for these factors using the language of existing frameworks such as the Consolidated Framework for Implementation Research (CFIR) [17]. Our analytic approach emphasized being inductively open to what the "open-ended" questions might generate. Some identified factors (such as "Veteran preferences and needs") neatly align with and confirm the centrality of CFIR constructs (in this case, "patient needs and preferences"). Others, such as "perception of scope," which we identified as an influential factor, do not correspond to any one CFIR construct, touching instead on the intersection of personal characteristics as well as organizational climate. These more inductive findings thus stimulate discussion around refining and using theoretical models for practical purposes. Informed as it is by insights from implementing clinicians, our work helps set the stage for informed efforts to promote greater implementation of evidence-based programs such as PTM in the VA.

## Available resources

As is common in implementation of new programs [19, 20], staffing shortages, lack of space, and limited availability of materials such as patient workbooks were identified as barriers for PTM implementation. Tuepker et al [16] found that administrative time and/or administrative support facilitated PTM implementation, and that lack of this resource was a barrier to implementation. Alternatives to PTM group sessions, including individual care and telehealth delivery, have good evidence of efficacy [11]. Encouraging use of these alternatives to group sessions may help overcome the issue of finding space for groups while providing more options to align with patient preferences. Additionally, telehealth delivery of PTM allows for flexibility during unexpected situations, such as the recent global pandemic, which may be of increasing importance in both the immediate and long-term future [21].

## Service collaboration

Although the traditional home of tinnitus care is audiology, both audiology and mental health expertise are critical to the successful implementation of PTM. A much larger proportion of audiology respondents than mental health respondents reported provision of PTM at their sites. It is likely that PTM is provided at some sites with little or no involvement from mental health. As PTM is typically initiated in audiology, it is often left up to audiologists to make efforts to connect with mental health. However, there is no established pathway or clear guidance for audiology partnering with mental health. As mental health clinicians may not understand their role with respect to provision of tinnitus care and are already stretched thin, these efforts at establishing collaboration may not be successful. Further research is needed to understand how audiology and mental health collaborate to provide PTM at sites that have successfully implemented the program. Stories describing successful collaboration between audiology and mental health can help inspire those starting PTM.

One survey respondent suggested that PTM could be done in primary care within Primary Care-Mental Health Integration (PCMHI) services, which are common throughout the VA. PCMHI programs assist patients with a wide range of behavioral health assessments, brief interventions, and referrals to higher levels of mental health care as needed. Because PCMHI providers are familiar with the role of a mental health provider in helping patients to cope with physical problems such as pain, PCMHI may well be a way to help Veterans in coping with tinnitus. The authors are aware of some VA sites experimenting with adapting PTM for use within PCMHI; however, further research is needed to develop and test a well-informed

PCMHI version of PTM. In the meantime, PCMHI providers could be educated about PTM so that they can help to connect patients to these services where available.

Due to the recent COVID-19 global pandemic, use of telehealth services has become more predominant. It is possible that increased use of telemental health services may serve to reduce perceived barriers to contributing to a Tele-PTM program. A mental health provider who is already providing telehealth services will not have the barrier of lack of space or inability to coordinate the use of space with audiology—they can remain in the same space and use the same telehealth technology used with other patients. Also, when seeking out a mental health provider to collaborate with in providing Tele-PTM, there are fewer restrictions in terms of location of provider, as telehealth may be provided from other locations.

## Prioritization

Mental health providers treat patients with many serious conditions, and tinnitus likely is not a top priority for many. Prioritization can be linked to availability of resources for particular programs, and also can be linked to buy-in from supervisors or senior management on where clinicians should focus their attention. Tuepker et al [16] provided examples of ways supervisors can signal their buy-in to PTM implementation, including mandatory training and dedicated provider time. Improvement in the availability of resources (for example) may also benefit prioritization.

A large proportion of tinnitus patients have comorbid mental health diagnoses [7]. One pathway to improve prioritization of care for tinnitus within mental health could involve education about the impact of care for tinnitus on comorbid mental health conditions. There is some evidence that symptoms of anxiety and depression improve when people are provided with care for tinnitus, even when that care is not focused on anxiety or depression [22]. For example, in the prior RCTs of PTM, statistically significant improvements in anxiety and depression symptoms were documented in addition to tinnitus-related quality of life [11].

## Veteran preferences and needs

Audiologist survey responses noted that Veterans may object to traveling to the VA for multiple PTM group sessions due to the logistical challenges involved (e.g., distance of travel required, number of appointments, parking, time off work, etc.). In addition to these stated challenges, other events (e.g., the COVID-19 global pandemic) can create barriers to providing in-person care, and to care in groups. Availability of telehealth options can help sites overcome some of these issues, as travel is not required to receive care remotely, and individual appointments can be made with more flexibility around the Veteran's schedule compared to group sessions on a pre-set schedule. Henry et al [11] found that one-on-one telephone administration of PTM resulted in better attendance rates and better outcomes when compared to a study of in-person group PTM [3]. Mental health survey responses suggested that Veterans may not be aware that their mental health provider could help with their tinnitus, and may not bring up tinnitus during mental health appointments for that reason. These providers clearly value the perspective of Veterans, though it is unknown whether their speculation correctly represents Veterans. Input collected directly from Veterans is required to ensure VA programs meet their needs.

## Clinician training

The majority of audiology survey respondents and well over half of mental health survey respondents indicated an interest in receiving training in tinnitus management. Given the stated preferences for duration of training, audiologists may be more receptive to longer

training opportunities (i.e., 5–20 hours). Nearly a third of mental health respondents were willing to invest in equally long training opportunities; however, more than one third of mental health respondents preferred training of 3 hours or fewer. It may be helpful to offer both shorter and longer versions of training opportunities to meet the needs and preferences of both audiology and mental health providers.

Based on survey results, it appears there may be reluctance from audiologists and mental health providers to engage in training that requires travel. Although audio- or video-recording sessions would allow for training and ongoing mentorship with no travel for provider or trainer, these options were not popular with either set of providers. Assessing willingness to travel when developing educational opportunities for clinicians may be important. Write-in suggestions from both disciplines favored more mentoring support (e.g., going over difficult cases by phone) than direct oversight and feedback. Training and ongoing mentorship by telephone may be palatable for both trainers and trainees. Such 'hub-and-spoke' or 'train-the-trainer' models of dissemination have been used to implement other evidence-based practices across the VA nationally [23–28]. Similarly, a centralized tinnitus education and resource hub, into which clinicians could call for support as needed, could be an effective delivery model to meet the education and mentorship needs of audiologists and mental health providers who work with patients with tinnitus.

## Knowledge of evidence-based options

Most VA mental health providers are trained in CBT as there is a strong focus on providing this evidence-based practice for relevant mental health disorders. Although mental health providers regularly use CBT for various conditions, they are not necessarily aware that there is good evidence for the use of CBT for tinnitus. There is also support in the literature, though not as robust, for other mental health strategies with tinnitus such as Acceptance and Commitment Therapy (ACT) [29]. These methods are not unknown to mental health providers; rather, the barrier is connecting these particular methods to tinnitus management specifically. Outreach is needed to help raise awareness and make the connection between existing evidence-based practices and their application to tinnitus care. Mental health providers already offering care with patients coping with other chronic health problems (pain, cancer, etc.) would be obvious initial targets for such training.

Our survey indicated that audiologists perceive a fair amount of inconsistency in their field regarding awareness and use of evidence-based methods for tinnitus. Audiologists predominantly reported that they can and should provide care for tinnitus. However, to date, there is no evidence-based therapy for tinnitus that involves audiology alone. In interviews with VA audiologists and mental health providers conducted by Tuepker et al [16], providers identified as clinical champions of PTM tended to be audiologists. Thus, although audiologists are willing to champion PTM, they cannot provide this evidence-based care for tinnitus without mental health providers to deliver the CBT portion of the care. As discussed further below, future research that examines audiologists' use of key elements of CBT to help patients with tinnitus could change this dynamic. Until this possibility is explored further, it may be most helpful to support audiologists in championing evidence-based practices, while simultaneously supporting them in efforts to collaborate with mental health providers.

## Perception of scope

Most audiologists view tinnitus as within their scope of practice, though they recognize that comorbidities such as anxiety and depression lie outside their scope. However, mental health providers tend not to see tinnitus as part of their scope of practice, which aligns with the

reported lack of knowledge of evidence-based options for tinnitus among mental health providers. Drawing parallels between tinnitus and chronic pain [30, 31], which is recognized as within the scope of mental health, can aid the argument that tinnitus is worthy of attention from mental health as well. As discussed above, CBT is an effective approach for improving quality of life for people with chronic conditions, and there is evidence supporting use of CBT in treating both tinnitus and chronic pain [30, 31]. Increased collaboration with audiology, and prioritization of tinnitus within mental health, would also underscore the perception that tinnitus falls within the scope of mental health.

## Audiologists and CBT

When mental health providers are not engaged in providing care for tinnitus within a PTM program, audiologists will sometimes consider conducting the CBT portion of PTM themselves. However, providing CBT is not typically considered to be within an audiologist's scope of practice. Mental health providers receive extensive training and supervision to become proficient in CBT, and most audiologists do not have opportunities for equally stringent training and supervision. More importantly, mental health providers have a breadth and depth of knowledge that is relevant to people struggling with any chronic condition that audiologists do not have. It is in the best interest of patients struggling with tinnitus to have access to a mental health provider as part of the team of clinicians providing care for tinnitus.

When it is not possible to include a mental health provider in providing tinnitus care, audiologists may leverage some key elements of CBT without going outside their scope of practice. Relaxation techniques and planning pleasant activities are mind-body practices that overlap with CBT. The VA system is advancing a Whole Health program [32, 33] that endorses VA health care providers of any discipline teaching mind-body practices to their patients. In accordance with these guidelines, it would be acceptable for audiologists to teach deep breathing, imagery, and other mind-body approaches to coping. It is possible that these mind-body approaches, along with the sound therapy elements of PTM coping skills education, could be delivered to patients with tinnitus entirely by audiologists. Future research to evaluate the efficacy and effectiveness of this option is warranted. Audiologists teaching mind-body approaches to coping with tinnitus does not eliminate the need for behavioral health involvement; it may, however, improve the quality of care provided by audiologists when they are unable to partner with behavioral health.

## Comparison to other research

Several other studies have examined factors affecting implementation of tinnitus strategies. Cima [34] also identified the availability of resources as a factor, which impacted the involvement of psychologists in tinnitus care and the presence of multidisciplinary care teams. They also identified lack of clinician knowledge and lack of consensus on treatment options as barriers, which relate to our themes of clinician training and knowledge of evidence-based options. Martin et al [35] noted the impact of using evidence-based intervention and providing high-level training on the long-term sustainability of a program. Beck et al [36] identified close collaboration between an audiologist and a psychologist as an important factor in providing tinnitus care (specifically PTM). They attributed the success of the program, in part, to the transdisciplinary approach of the clinicians involved, whereby the boundaries between disciplines are blurred. This finding illustrates the importance of our theme of service collaboration, and its importance to a PTM program. Additionally, Beck et al identified the lack of administrative time and support as a challenge to their program, which fits under our theme of availability of resources. Finally, Beck et al paid close attention to Veteran needs by adapting

the program to better fit Veteran preferences and noting that participation may have been limited at one site due to Veterans' reluctance to travel long distances for multiple group sessions. Further, they noted that telehealth options would increase the potential reach of their program. These views align well with our study findings. Another study of clinical implementation of PTM by Edmonds et al [37] highlighted the importance of interdisciplinary care, but also described the challenge of achieving this model of care (one site did not have mental health involvement in the PTM program). Edmonds et al also reported successful provision of PTM via video conferencing.

## Limitations

This survey was conducted in 2015, and results reported here represent an estimation of the clinical provision of tinnitus services and PTM at that time. It is likely that changes have occurred in these areas since the administration of the survey. However, as no large-scale PTM implementation efforts have occurred since this survey, it is likely these data are still largely representative of VA clinical practices for tinnitus.

As with any voluntary survey, response bias may influence the results presented here. The participation rate was approximately 60% for audiology and 46% for mental health and, although this is similar to response rates of other studies involving VA clinicians [38–40], these respondents may not be representative of all eligible providers. Those electing to respond to a survey about tinnitus may be more likely to have strong feelings (either positive or negative) about tinnitus, providing care for tinnitus, or tinnitus patients. In particular, responses among audiologists may not be contrastable to those among mental health providers, as there could be systematic differences between those who participated from each discipline and the sites at which they work. Our purposive sampling strategy attempted to gather data from those who were most "hands on" in their sites' tinnitus programs, but in taking this approach we did not gather data from administrators or others whose decisions about service management may also critically influence the implementation of PTM.

Additionally, responses between the disciplines could not be linked between sites due to little overlap. Dyadic surveys or interviews, where clinicians who work together across disciplinary lines provide joint responses, could have yielded a more comprehensive, integrated picture of each site's program; however, these approaches were not feasible due to the additional clinician burden they would impose. Future work that can assess the consistency of responses between audiologists and mental health providers from the same sites would help inform efforts to increase collaboration between these two disciplines.

## Future directions

The results of this study, though informative, are now five years old. Our next effort includes conducting a follow-up survey to measure potential changes in PTM implementation in the intervening years. This follow-up survey is a joint effort between the VA and the Department of Defense (DoD), and will collect responses from audiologists, mental health providers, ear, nose, and throat physicians, and primary care physicians from both VA and DoD. The content of the follow-up survey is influenced by the results of the first survey reported in this manuscript. New training materials continue to be developed, and a series of online courses has recently been launched for both VA and DoD providers. Partnerships with VA health initiatives such as PCMHI, the Whole Health movement, and Clinical Resource Hubs will continue to be explored. Finally, as the landscape of care continues to change with the expansion of telehealth options, in particular the advent of telehealth services into the home, the focus will be on Tele-PTM to allow provision of PTM with minimal use of clinical space, and little to no

travel for Veterans. Implementing Tele-PTM will require attention to access for disadvantaged groups (poor, computer challenged, rural, deaf, visually impaired, etc.) due to the need for devices and access to a reliable internet signal in a private location. The VA, for example, provides a tablet device and a paid data plan as needed to ensure appropriate access. Investing in Tele-PTM also provides for a relatively seamless transition to remote care when situations such as global pandemics preclude the provision of in-person PTM.

## Conclusion

This study was the first to systematically examine the provision of tinnitus services across VA sites with the potential to provide evidence-based PTM (i.e., those with both audiology and mental health services). Results suggest wide variation in services provided, a need for greater engagement of mental health providers in tinnitus care, and an interest among both audiologists and mental health providers in receiving tinnitus-related training. Future efforts should focus on improving understanding among mental health providers of their potential role in tinnitus management, enhancing coordination of tinnitus-related care between audiology and mental health, collecting empirical data on Veterans' needs and interests, and examining the impact of audiologists teaching mind-body coping strategies that overlap with CBT elements when mental health is unavailable. Building on the success of providing PTM by telephone, the use of telehealth options to overcome location and resource challenges will become increasingly vital in the future.

## Supporting information

**S1 Appendix. Audiology survey.**
(PDF)

**S2 Appendix. Mental health survey.**
(PDF)

**S1 Dataset. Survey data.** Responses to closed-ended survey questions used in analyses. Data are presented in a randomized order. Some variables were recategorized to take into account write-in responses when there was an 'Other' option. The recategorized version is included in these cases. PTM Category was determined by the study team based on other survey responses.
(XLSX)

**S2 Dataset. Qualitative data.** Responses to open-ended survey questions used in qualitative analysis. Blank (i.e. missing) responses have been excluded. Responses for each survey item are presented in a randomized order, organized by discipline. Text that has been changed in order to de-identify the respondent is indicated in [square brackets], any text that has been removed for de-identification purposes is indicated by square brackets with an ellipsis [. . .]. Typos in responses have been indicated by (sic).
(DOCX)

## Author Contributions

**Conceptualization:** Tara L. Zaugg, Emily J. Thielman, Kathleen F. Carlson, Anaïs Tuepker, Christine Elnitsky, Summer Newell, Christine Kaelin, James A. Henry.

**Data curation:** Emily J. Thielman, Christie Choma.

**Formal analysis:** Tara L. Zaugg, Emily J. Thielman, Anaïs Tuepker, Christine Elnitsky, Summer Newell.

**Funding acquisition:** James A. Henry.

**Investigation:** Emily J. Thielman, Christine Elnitsky, James A. Henry.

**Methodology:** Tara L. Zaugg, Emily J. Thielman, Kathleen F. Carlson, Anaïs Tuepker, Christine Elnitsky, Summer Newell, Christine Kaelin.

**Project administration:** Emily J. Thielman, Christine Kaelin.

**Supervision:** James A. Henry.

**Validation:** Christine Elnitsky, Christie Choma.

**Visualization:** Emily J. Thielman, Anaïs Tuepker.

**Writing – original draft:** Tara L. Zaugg, Emily J. Thielman, Christine Elnitsky, Karen L. Drummond, Christine Kaelin, Christie Choma, James A. Henry.

**Writing – review & editing:** Tara L. Zaugg, Emily J. Thielman, Kathleen F. Carlson, Anaïs Tuepker, Christine Elnitsky, Karen L. Drummond, Caroline J. Schmidt, Summer Newell, Christine Kaelin, James A. Henry.

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
