## [Decision Letter · Decision Letter 0]

15 May 2020

PONE-D-20-09484

Factors Affecting the Implementation of Evidence-based Progressive Tinnitus Management in Department of Veterans Affairs Medical Centers

PLOS ONE

Dear Dr. Zaugg,

Thank you for submitting your manuscript to PLOS ONE. After careful consideration, we feel that it has merit but does not fully meet PLOS ONE’s publication criteria as it currently stands. Therefore, we invite you to submit a revised version of the manuscript that addresses the points raised during the review process.

We would appreciate receiving your revised manuscript by Jun 29 2020 11:59PM. To enhance the reproducibility of your results, we recommend that if applicable you deposit your laboratory protocols in protocols.io, where a protocol can be assigned its own identifier (DOI) such that it can be cited independently in the future. For instructions see: http://journals.plos.org/plosone/s/submission-guidelines#loc-laboratory-protocols

We look forward to receiving your revised manuscript.

Kind regards,

Vinaya Manchaiah, AuD, MBA, PhD

Academic Editor

PLOS ONE

Journal Requirements:

"The authors have read the journal’s policy and have the following conflicts: JH, TZ, and CS were part of the research team that originally developed Progressive Tinnitus

Management. TZ, ET, KC, AT, KD, SN, CK, CC, and JH are currently affiliated with the

Department of Veterans Affairs. The views expressed in this article are those of the

authors and do not necessarily reflect the position or policy of the Department of

Veterans Affairs or the United States government."

Reviewers' comments:

Reviewer's Responses to Questions

**Comments to the Author**

1. Is the manuscript technically sound, and do the data support the conclusions?

Reviewer #1: Partly

Reviewer #2: Yes

2. Has the statistical analysis been performed appropriately and rigorously? 

Reviewer #1: No

Reviewer #2: Yes

3. Have the authors made all data underlying the findings in their manuscript fully available?

Reviewer #1: Yes

Reviewer #2: Yes

4. Is the manuscript presented in an intelligible fashion and written in standard English?

Reviewer #1: Yes

Reviewer #2: Yes

5. Review Comments to the Author

Reviewer #1: Factors Affecting the Implementation of Evidence-based Progressive Tinnitus Management in Department of Veterans Affairs Medical Centers

GENERAL COMMENT

This is a well written paper with a clear rationale. PTM is and evidence-based approach including multidisciplinary aspects and one of the few examples of a step-care approach. Although uptake is indorse in the VA has been slow and inconsistent. This is a valid paper looking at factors that could be affecting this uptake. There are limitations regarding reporting of the survey development and data analysis methods that require attention before this paper will provide a valuable contribution to the scientific literature.

METHODS

1. There is very little detail provided about the development of the survey. This section could be expanded.

-E.g. how many people piloted it?

-What validation was done by the pilot panel?

-What adjustment had to be made?

2. Please provide more information about the questions themselves. I was not able to see Appendix A, but the paper itself needs and indication of how many closed ended questions, Likert Questions, Open ended questions.

3. The survey was conducted in 2015. This seems quite a long time ago and means the results may be very different at the present time. Could some explanation be provided for this time delay and how to account for this. It was mentioned as a limitation, but no explanations provide for this lag.

4. The data anlaysis section is very limited. As not enough information is provided about the types of questions it is not clear what analyses would be useful. Were there Likert scaled questions? The only information provided is that frequencies were recorded. Further analysis is desirable.

- For instance were there differences in responses between the audiologist and MH clinicians? It seems as though this data has not thoroughly been explored at all

- Comparisons between Full PTM and partial PTM sites could also be made

5. The qualitative analysis could also be expanded.

- It is not clear what theoretical framework was used

- Was coverage of the theme calculated?

- Which themes were reported and not reported?

- What was the inter-rated reliability? There were many meetings to resolved differences- how many differences were there?

RESULTS

1. More information is required regarding the reach. There were 144 sites targeted, not sure if you can expand on how many mental health professionals and audiologists were targeted? What percentage responded? (87 audiologists, 66 mental health clinicians)

2. Some statistical analysis is desired to at least compare responses between the categories. Was there a difference in these responses between the above mentioned groups or partial and full PTM sites? Comparisons between groups can be made to help make more sense of the data

3. The results are very hard to interpret. The use of Figures is desired and would be very helpful. There figures could compare audiology and MH responses or Full and Partial responses.

4. The tables are really hard to draw conclusions from. They are just presenting the data as is with no comparison between the groups or knowing which were providing full or partial PTM. More than just the descriptive data is required.

5. Qualitative analysis- what was the coverage for each theme and each type of professional?

- Which themes were present from sites with full PTM and which for partial PTM

-Table 4 would be good to add how many There is no indication of coverage for these themes

- It would be helpful to break down the subthemes. How many people mentioned each of these subthemes. This will indicate where bigger barriers like

DISCUSSION

The discussion would benefit from a more critical discussion of the methods used and results (when expanded on during analysis).

-Some account of how these compare with the implementation of other tinnitus approaches is required

- Outline what has been gained from doing the study regarding next steps to take to improve implementation of this program and overcome the identified barriers.

- As mentioned in the methods, it is quite concerning that this survey was done such a long time ago and only reported upon now. It does cast doubts on the validity for the present situation.

ABSTRACT

Once more work has been done the abstract can be updated. The results and conclusions section hardly provide any information. It provides there impression that these is not that much to report when the conclusion states the study aim again.

Reviewer #2: This MS offers a straightforward presentation of findings from a survey offered to practitioners re: delivery of progressive tinnitus management (PTM) at VA hospitals. The findings provide information regarding PTM provision, and it is clear from the dataset that the intervention strategy is not practiced in a uniform manner across sites. The authors capture an unfortunate irony: the data in this MS make it clear that MH providers have the tools, but not particularly the interest/perceived scope of practice, to provide tinnitus management. At the same time, audiologists have the interest/perceived scope, but not the tools, for the same end. With this in mind, the authors could more reasonably cast a portion of the results as justification to support audiologists’ facility w/ CBT elements germane to tinnitus management. This notion may be outside the scope of this paper, but as an option, in this reviewer’s opinion, it merits mention. Stated another way, while the authors are to be commended for accurately reporting PTM’s encouragement of interprofessional care for bothersome tinnitus, their data suggest that such practice rarely occurs even in a system with designated opportunities and priorities for such practice (i.e., the VA). When viewed through the lens of perceived scope of practice, it would appear inappropriate to advocate for audiologists performing CBT. However audiologists, and care providers in dozens of other disciplines, employ tenets of CBT routinely. The authors correctly cite the important MH effects of bothersome tinnitus to propose that MH providers could be encouraged to take a greater interest in tinnitus. However their results also confirm that it is audiologists who are more likely to complete such training; why not, in that case, advocate more forcefully for training of audiologists?

Additional comments:

Line 61: if this passage refers to the sound of tinnitus, suggest changing “the tinnitus itself” to “the tinnitus sound”

Line 66: why quote marks around cure?

Line 76: audiologists counsel as well as the other items mentioned

Line 175: is the lack of overlapping site codes a concern? Perhaps specify that it affirms the lack of interdisciplinary care that PTM is intended to improve?

Line 281 and elsewhere: would be reasonable to include statements to the effect that telephone/telemed delivery of PTM and training for PTM may become more important in the immediate, and long-term future

Line 362: prefer change from “tinnitus” to “tinnitus management”

Lines 372-385: agree w/ the statement, however this reviewer suggests that another option could be included: “ …championing EB practices, improving audiologists’ use of CBT tenets in their routine tinnitus management, while simultaneously…” The link to chronic pain is well-considered and important, but do the authors believe that illustrating similarities between chronic pain and tinnitus would help recruit MH providers to take on patients w/ bothersome tinnitus? Chronic pain still presents a major challenge for providers; perhaps the notion that adding tinnitus to that load would repel providers rather than encourage providers requires consideration.

Line 412: perhaps reference the success reported (earlier in the MS) re: the telephone delivery’s success in addition to the other items comprising the conclusion.

6. PLOS authors have the option to publish the peer review history of their article (what does this mean?). If published, this will include your full peer review and any attached files.

Reviewer #1: No

Reviewer #2: No

---

## [Author Response · Author response to Decision Letter 0]

1 Oct 2020

Comment 1. Please ensure that your manuscript meets PLOS ONE's style requirements, including those for file naming. The PLOS ONE style templates can be found at

and

Response 1. Thank you for providing these links to the style templates, they were very helpful. The manuscript has been adjusted to meet these guidelines. 

Comment 2. Please include additional information regarding the survey or questionnaire used in the study and ensure that you have provided sufficient details that others could replicate the analyses. For instance, if you developed a questionnaire as part of this study and it is not under a copyright more restrictive than CC-BY, please include a copy, in both the original language and English, as Supporting Information. 

Response 2. We thank the editors/reviewers for these helpful suggestions. In the revised manuscript, we have added details about the development of our survey questionnaires and the analysis of survey results. We are also now including both full survey questionnaires as supporting information with this resubmission. 

Comment 3. We note that you have indicated that data from this study are available upon request. PLOS only allows data to be available upon request if there are legal or ethical restrictions on sharing data publicly. For more information on unacceptable data access restrictions, please see http://journals.plos.org/plosone/s/data-availability#loc-unacceptable-data-access-restrictions.

Response 3. We have further investigated our ability to release the qualitative data that was analyzed for this article. We have taken steps to de-identify the data, and have been told by our privacy office that it may be shared in this de-identified form. With this qualitative data added as supplementary information, we will now be making the complete dataset available. 

Comment 4. Thank you for stating the following in the Competing Interests section: "The authors have read the journal’s policy and have the following conflicts: JH, TZ, and CS were part of the research team that originally developed Progressive Tinnitus Management. TZ, ET, KC, AT, KD, SN, CK, CC, and JH are currently affiliated with the Department of Veterans Affairs. The views expressed in this article are those of the authors and do not necessarily reflect the position or policy of the Department of Veterans Affairs or the United States government."

Response 4. In our original submission, we included this statement: "The authors have read the journal’s policy and have the following conflicts: JH, TZ, and CS were part of the research team that originally developed Progressive Tinnitus Management. TZ, ET, KC, AT, KD, SN, CK, CC, and JH are currently affiliated with the Department of Veterans Affairs. The views expressed in this article are those of the authors and do not necessarily reflect the position or policy of the Department of Veterans Affairs or the United States government." The disclosure does not alter our adherence to PLOS ONE policies on sharing data and materials.

Comment 5. PLOS requires an ORCID iD for the corresponding author in Editorial Manager on papers submitted after December 6th, 2016. Please ensure that you have an ORCID iD and that it is validated in Editorial Manager. To do this, go to ‘Update my Information’ (in the upper left-hand corner of the main menu), and click on the Fetch/Validate link next to the ORCID field. This will take you to the ORCID site and allow you to create a new iD or authenticate a pre-existing iD in Editorial Manager. Please see the following video for instructions on linking an ORCID iD to your Editorial Manager account: https://www.youtube.com/watch?v=_xcclfuvtxQ

Response 5. The submitting author’s ORCID ID has been added to the submission. 

Comment 6. Please include captions for your Supporting Information files at the end of your manuscript, and update any in-text citations to match accordingly. Please see our Supporting Information guidelines for more information: http://journals.plos.org/plosone/s/supporting-information.

Response 6. Thank you. We have adjusted our file names, in-text citations, and added captions according to these guidelines. 

Reviewer #1

GENERAL COMMENT

This is a well written paper with a clear rationale. PTM is and evidence-based approach including multidisciplinary aspects and one of the few examples of a step-care approach. Although uptake is indorse in the VA has been slow and inconsistent. This is a valid paper looking at factors that could be affecting this uptake. There are limitations regarding reporting of the survey development and data analysis methods that require attention before this paper will provide a valuable contribution to the scientific literature.

METHODS

Comment 7. There is very little detail provided about the development of the survey. This section could be expanded.

-E.g. how many people piloted it?

-What validation was done by the pilot panel?

-What adjustment had to be made?

Response 7. Additional details about the survey development process have been added to the Methods.

Comment 8. Please provide more information about the questions themselves. I was not able to see Appendix A, but the paper itself needs and indication of how many closed ended questions, Likert Questions, Open ended questions.

Response 8. We appreciate the reviewer’s suggestion and have added more information about the survey questions in our revised manuscript (second paragraph under the heading Survey Development and Data Collection in the Methods section). Additionally, we have now included both full survey questionnaires as Appendix A in our resubmission. We hope that this adequately addresses the reviewer’s concerns. 

Comment 9. The survey was conducted in 2015. This seems quite a long time ago and means the results may be very different at the present time. Could some explanation be provided for this time delay and how to account for this. It was mentioned as a limitation, but no explanations provide for this lag.

Response 9. We understand the reviewer’s concern and feel we can justify the time lag. This survey was originally meant to inform another, more direct implementation effort. However, the funding mechanism for that follow-up project was discontinued before we could apply for it. With the original plan for the project somewhat derailed, analysis and publication of these survey data were deprioritized while alternate sources of funding were secured. Although five years have elapsed, we strongly believe the data presented in this manuscript continue to be an important addition to the literature. VA policy and practice have continued to recommend PTM without substantial investments in facilitation of the implementation process; thus, many aspects of the 2015 context remain salient today, and can be used to inform ongoing implementation improvement efforts. 

Comment 10. The data analysis section is very limited. As not enough information is provided about the types of questions it is not clear what analyses would be useful. Were there Likert scaled questions? The only information provided is that frequencies were recorded. Further analysis is desirable.

- For instance were there differences in responses between the audiologist and MH clinicians? It seems as though this data has not thoroughly been explored at all

- Comparisons between Full PTM and partial PTM sites could also be made

Response 10. Thank you for this comment and these suggestions. We have revisited and considered additional analyses that could be added to the manuscript. The potential comparison of audiologist and mental health clinician responses is a compelling idea and one that we would have liked to pursue. However, we are unable to make direct comparisons between clinician types because of our inability to identify whether surveys were returned from the same site(s). Overall, though, the data presented in our tables suggest that audiologists were more in tune with tinnitus management than mental health clinicians, which is what we expected to see. Regarding the comparison between levels of PTM (full, partial, and no), we examined crosstabulations of data and found that this analysis was limited by the number of low or zero count cells, particularly in the mental health survey data. After considering the reviewer’s suggestions, and where data were sufficient to allow, we have added additional results to the manuscript to more thoroughly explore all of the available data. 

Comment 11. The qualitative analysis could also be expanded.

- It is not clear what theoretical framework was used

- Was coverage of the theme calculated?

- Which themes were reported and not reported?

- What was the inter-rated reliability? There were many meetings to resolved differences- how many differences were there?

Response 11. We thank the reviewer for encouraging a more detailed description of our qualitative analysis. We used a primarily inductive approach to qualitative analysis whilst recognizing that themes were likely to be influenced by the a priori domains from the general implementation science perspective that guided the study overall. We did not calculate coverage of themes since variation of themes was of more central importance to our research question (all themes were reported); additionally, coverage calculations may give an impression of ranked importance which is not justified by data collected through open-ended responses. We did, however, note differences in the presence or nuance of themes as they arose in the responses of MH versus Audiology providers; this is noted in Table 4. Though frequently applied to qualitative health research in the past, inter-rater reliability is increasingly viewed as more methodologically appropriate for quantitative studies, with methods such as multi-coder review and iterative, regular team discussion to refine theme development accepted as more appropriate tools for qualitative analysis (see RS Barbour’s “Checklists for improving rigour in qualitative research: a case of the tail wagging the dog?” BMJ 2001: 322: 1115-7). We have edited the text to clarify the reasons for our iterative analytic process (in which “meetings to resolve differences” are an expected and necessary part of the analytic process, even when differences in interpretation are minor, as was the case here) (second paragraph under the heading Quantitative and Qualitative Analyses).

RESULTS

Comment 12. More information is required regarding the reach. There were 144 sites targeted, not sure if you can expand on how many mental health professionals and audiologists were targeted? What percentage responded? (87 audiologists, 66 mental health clinicians)

Response 12. The survey was sent to a list of 144 audiology service chiefs and 144 mental health service chiefs representing each VA facility with both of these services. Our aim was to collect one audiology and one mental health response from each site. The instructions accompanying the survey link asked the service chief to either complete the survey themselves, or to give it to a clinician knowledgeable about tinnitus clinical services to complete. Our receipt of 87 audiology responses and 66 mental health responses roughly represents response rates of 60.4% and 45.8%, respectively. This rate of response is similar to other projects in which VA staff have been surveyed with response rates between 30% and 60% (see references below). We now include this additional detail in the revised manuscript. 

McIntosh N, Meterko M, Burgess JF, et al. Organizational predictors of coordination in inpatient medicine. Health Care Management Review, 2014a;39(4):279-92, 2014.

McIntosh N, Burgess JF, Meterko M, et al. Impact of provider coordination on nurse and physician perceptions of patient care quality. Journal of Nursing Care and Quality, 2014b;29(3):269-79, 2014.

Pogoda TK, Carlson KF, Gormley KE, Resnick SG. Supported Employment for Veterans with Traumatic Brain Injury: Provider Perspectives. Archives of Physical Medicine and Rehabilitation, special issue on Community Reintegration, Participation, and Employment Issues in Veterans and Service Members with Traumatic Brain Injury; 99(2 Suppl): S14-S22, 2018.

Comment 13. Some statistical analysis is desired to at least compare responses between the categories. Was there a difference in these responses between the above-mentioned groups or partial and full PTM sites? Comparisons between groups can be made to help make more sense of the data.

Response 13. As stated above in our response to Comment 10, we examined crosstabulations of data using the PTM categories and found that we were hampered by the number of low or zero count cells. Where possible, we have added results to the manuscript to show that this avenue of analysis has been explored.

Comment 14. The results are very hard to interpret. The use of Figures is desired and would be very helpful. There figures could compare audiology and MH responses or Full and Partial responses.

Response 14. We appreciate the reviewer’s suggestion. However, we have revisited and discussed the use of figures and we feel that the most straightforward presentation of the results, in this case, is in tables. We are being cautious to not present visual representations that might imply meaningful trends in cases where such trends are not actually there. Additionally, the small N observed in the Partial PTM category makes comparisons of full and partial PTM responses difficult.

Comment 15. The tables are really hard to draw conclusions from. They are just presenting the data as is with no comparison between the groups or knowing which were providing full or partial PTM. More than just the descriptive data is required.

Response 15. As noted in response to comment 10, when stratifying the data by PTM category, the table cells were quite sparse, with numerous empty cells. This presentation of the data, we felt, was more difficult to interpret than just showing the results by survey type. We would have liked to present our results grouped by PTM category, but due to the relatively low number of responses we received across these categories, the data did not support such comparisons. 

Comment 16. Qualitative analysis- what was the coverage for each theme and each type of professional?

- Which themes were present from sites with full PTM and which for partial PTM

-Table 4 would be good to add how many There is no indication of coverage for these themes

- It would be helpful to break down the subthemes. How many people mentioned each of these subthemes. This will indicate where bigger barriers like

Response 16. Please see comment 11 for response to “coverage of themes.”

DISCUSSION

Comment 17. The discussion would benefit from a more critical discussion of the methods used and results (when expanded on during analysis).

Response 17. We appreciate this suggestion. As described above, we now include more detail about the survey methods and data analysis in the Methods section. We also have added reflections on how both our coding methods and our findings relate to a frequently used implementation science framework, in the first paragraph of the Discussion. We have added additional reflections on limitations of our sampling strategy and analysis in the Limitations section.

Comment 18. Some account of how these compare with the implementation of other tinnitus approaches is required

Response 18. Thank you for this suggestion. We have added a section to the Discussion called Comparison to other Research, which describes similar themes found in other studies of implementation of tinnitus strategies. Additionally, we have included a reference to implementation of PTM within the Discussion, under the heading Available Resources there is a description of barriers to implementing PTM that are common to implementing any new program in a clinical setting. 

Comment 19. Outline what has been gained from doing the study regarding next steps to take to improve implementation of this program and overcome the identified barriers.

Response 19. We added a section to the discussion with the heading Future Directions. It describes next steps influenced by the work described in the manuscript. 

Comment 20. As mentioned in the methods, it is quite concerning that this survey was done such a long time ago and only reported upon now. It does cast doubts on the validity for the present situation.

Response 20. We can understand the reviewer’s concern. It is indeed likely that some things have changed in 5 years; however, VA healthcare system is big and tends to change slowly. In fact, we have been contacted by VA clinicians with questions and anecdotal reports of barriers experienced/overcome in the intervening years. Many issues identified by this survey continue to come up in these communications with VA clinicians. Therefore, we believe the results of this study are valid and still relevant to contemporary issues in implementing tinnitus management services across a large, integrated healthcare system. 

ABSTRACT

Comment 21. Once more work has been done the abstract can be updated. The results and conclusions section hardly provide any information. It provides there impression that these is not that much to report when the conclusion states the study aim again.

Response 21.

Thank you for bringing this to our attention. The conclusions have been updated to address this concern. 

Reviewer #2

Comment 22. This MS offers a straightforward presentation of findings from a survey offered to practitioners re: delivery of progressive tinnitus management (PTM) at VA hospitals. The findings provide information regarding PTM provision, and it is clear from the dataset that the intervention strategy is not practiced in a uniform manner across sites. The authors capture an unfortunate irony: the data in this MS make it clear that MH providers have the tools, but not particularly the interest/perceived scope of practice, to provide tinnitus management. At the same time, audiologists have the interest/perceived scope, but not the tools, for the same end. With this in mind, the authors could more reasonably cast a portion of the results as justification to support audiologists’ facility w/ CBT elements germane to tinnitus management. This notion may be outside the scope of this paper, but as an option, in this reviewer’s opinion, it merits mention. Stated another way, while the authors are to be commended for accurately reporting PTM’s encouragement of interprofessional care for bothersome tinnitus, their data suggest that such practice rarely occurs even in a system with designated opportunities and priorities for such practice (i.e., the VA). When viewed through the lens of perceived scope of practice, it would appear inappropriate to advocate for audiologists performing CBT. However audiologists, and care providers in dozens of other disciplines, employ tenets of CBT routinely. The authors correctly cite the important MH effects of bothersome tinnitus to propose that MH providers could be encouraged to take a greater interest in tinnitus. However their results also confirm that it is audiologists who are more likely to complete such training; why not, in that case, advocate more forcefully for training of audiologists?

Response 22. We appreciate the reviewer’s suggestion. Having audiologists provide portions of CBT is an idea that comes up frequently but one that we are careful about referencing given the unique scopes of practice. We agree that it is worth briefly exploring in this manuscript given our findings. We have added text to the discussion section suggesting that audiologists could teach some elements of CBT (e.g., mind-body approaches) for working with tinnitus and that this solution may help overcome one of the primary barriers to the provision of tinnitus management to patients in need. This can be found in a new section called Audiologists and CBT.

Additional comments:

Comment 23. Line 61: if this passage refers to the sound of tinnitus, suggest changing “the tinnitus itself” to “the tinnitus sound”

Response 23. Thank you. This change has been made as suggested.

Comment 24. Line 66: why quote marks around cure?

Response 24. We appreciate this question. These quote marks have been removed from the revised manuscript since emphasis is not needed. 

Comment 25. Line 76: audiologists counsel as well as the other items mentioned

Response 25. We have reworded the sentence to call out the counseling aspect of audiologic care.

Comment 26. Line 175: is the lack of overlapping site codes a concern? Perhaps specify that it affirms the lack of interdisciplinary care that PTM is intended to improve?

Response 26. This is an interesting point and one that we considered in detail. However, we ultimately decided we cannot conclude that the lack of overlapping sites of our survey respondents reveals a lack of interdisciplinary care. For one, several audiology responses included a station number that is known to be erroneous (it is the mail code for audiology – common to many sites – rather than the site-specific station number). These responses could be from the same sites as some mental health responses, but there is no way to know for sure. In hindsight, this survey item should have had a validation rule to prevent erroneous responses and/or more clear explanation of what was being asked. We could also have received non-overlapping responses merely due to chance. There is also the possibility that the mental health chief we identified was not knowledgeable about who to send the survey to within their department. The clinical structure varied quite a bit among sites and some of the chiefs we identified may have been far removed from whichever clinicians might actually be providing tinnitus care. Those chiefs might have just ignored the survey if they did not know who to give it to.

Comment 27. Line 281 and elsewhere: would be reasonable to include statements to the effect that telephone/telemed delivery of PTM and training for PTM may become more important in the immediate, and long-term future

Response 27. We thank the reviewer for this excellent suggestion, and we agree wholeheartedly! Several statements have been added to this effect in the Discussion section. The changes can be found in the last sentence under the heading Available Resources, in the last paragraph of the section under the heading Service Collaboration, in the second sentence under the heading Veteran Preference and Needs, and the last sentence of the section under the heading Future Directions. 

Comment 28. Line 362: prefer change from “tinnitus” to “tinnitus management”

Response 28. This change has been made as suggested. 

Comment 29. Lines 372-385: agree w/ the statement, however this reviewer suggests that another option could be included: “ …championing EB practices, improving audiologists’ use of CBT tenets in their routine tinnitus management, while simultaneously…” 

The link to chronic pain is well-considered and important, but do the authors believe that illustrating similarities between chronic pain and tinnitus would help recruit MH providers to take on patients w/ bothersome tinnitus? Chronic pain still presents a major challenge for providers; perhaps the notion that adding tinnitus to that load would repel providers rather than encourage providers requires consideration.

Response 29.We considered the suggestion to add “improving audiologists’ use of CBT tenets in their routine tinnitus management,” however the intention for that section of the paper is to highlight the fact that there is no “evidence-based” care (at this point) that can be conducted by audiology only. The wording we previously provided did not make that clear enough. In the section referenced here, we have edited the wording to clarify that in order to be providing evidence-based care for tinnitus, mental health providers must be involved. As described above, we have included further description about the possibility of audiologists incorporating some of the tenets of CBT into their tinnitus services, though this would need to be empirically evaluated. 

The authors considered the point brought up by the reviewer regarding the possibility of repelling mental health providers by making parallels between providing care for tinnitus and providing care for chronic pain. The authors’ purpose for drawing parallels between providing care for tinnitus and providing care for chronic pain is to aid in the conceptualization of the role of mental health providers in providing care for tinnitus; drawing parallels with chronic pain accomplishes that. We agree that chronic pain presents a major challenge for providers. It is probably also true that providers who are repelled by working with people with chronic pain may also be repelled by working with people who struggle with tinnitus. Conversely, clinicians who are comfortable working with patients who have chronic pain are likely to also be comfortable working with people who have bothersome tinnitus. Our hope is that we will entice providers who are comfortable with the population and who feel well-equipped to work with them. 

Comment 30. Line 412: perhaps reference the success reported (earlier in the MS) re: the telephone delivery’s success in addition to the other items comprising the conclusion. 

Response 30. Thank you, for this suggestion. We have moved the referenced sentence to the Future Directions section, where it is now discussed more fully. The concluding line of the manuscript also references back to the telephone delivery of PTM described earlier.

---

## [Decision Letter · Decision Letter 1]

26 Oct 2020

Factors affecting the implementation of evidence-based Progressive Tinnitus Management in Department of Veterans Affairs Medical Centers

PONE-D-20-09484R1

Dear Dr. Zaugg,

We’re pleased to inform you that your manuscript has been judged scientifically suitable for publication and will be formally accepted for publication once it meets all outstanding technical requirements.

Kind regards,

Vinaya Manchaiah, AuD, MBA, PhD

Academic Editor

PLOS ONE

Additional Editor Comments (optional):

Reviewers' comments:

Reviewer's Responses to Questions

**Comments to the Author**

1. If the authors have adequately addressed your comments raised in a previous round of review and you feel that this manuscript is now acceptable for publication, you may indicate that here to bypass the “Comments to the Author” section, enter your conflict of interest statement in the “Confidential to Editor” section, and submit your "Accept" recommendation.

Reviewer #1: All comments have been addressed

Reviewer #2: All comments have been addressed

2. Is the manuscript technically sound, and do the data support the conclusions?

Reviewer #1: Yes

Reviewer #2: Yes

3. Has the statistical analysis been performed appropriately and rigorously? 

Reviewer #1: Yes

Reviewer #2: Yes

4. Have the authors made all data underlying the findings in their manuscript fully available?

Reviewer #1: Yes

Reviewer #2: Yes

5. Is the manuscript presented in an intelligible fashion and written in standard English?

Reviewer #1: Yes

Reviewer #2: Yes

6. Review Comments to the Author

Reviewer #1: Dear Authors

Thank you for addressing the comments. They have been sufficiently covered where possible and reasons have been provided where addressing the comments are not possible.

Reviewer #2: Thank you for the care taken in addressing reviewer concerns. The issue of audiologists and CBT provision was handled reasonably and adds to the MS's value.

7. PLOS authors have the option to publish the peer review history of their article (what does this mean?). If published, this will include your full peer review and any attached files.

Reviewer #1: No

Reviewer #2: No

---

## [Editor Report · Acceptance letter]

14 Dec 2020

PONE-D-20-09484R1 

Factors affecting the implementation of evidence-based Progressive Tinnitus Management in Department of Veterans Affairs Medical Centers 

Dear Dr. Zaugg:

I'm pleased to inform you that your manuscript has been deemed suitable for publication in PLOS ONE. Congratulations! Your manuscript is now with our production department. 

Kind regards, 

on behalf of

Dr. Vinaya Manchaiah 

Academic Editor

PLOS ONE